# Comparison of Diatoms and Dinoflagellates from Different Habitats as Sources of PUFAs

**DOI:** 10.3390/md17040233

**Published:** 2019-04-19

**Authors:** Elina Peltomaa, Heidi Hällfors, Sami J. Taipale

**Affiliations:** 1Faculty of Biological and Environmental Sciences, Ecosystems and Environment Research Programme, University of Helsinki, Niemenkatu 73, FI-15140 Lahti, Finland; 2Institute of Atmospheric and Earth System Research (INAR)/Forest Sciences, University of Helsinki, P.O. Box 27, FI-00014 Helsinki, Finland; 3Helsinki Institute of Sustainability Science (HELSUS), Yliopistonkatu 3, FI-00014 Helsinki, Finland; 4Finnish Environment Institute, Marine Research Centre, Marine Ecological Research Laboratory, Agnes Sjöbergin katu 2, FI-00790 Helsinki, Finland; heidi.hallfors@ymparisto.fi; 5Department of Biological and Environmental Science, University of Jyväskylä, P.O. Box 35 (YA), 40014 Jyväskylä, Finland

**Keywords:** microalgae, diatoms, dinoflagellates, marine, brackish, freshwater, polyunsaturated fatty acids, EPA, DHA

## Abstract

Recent studies have clearly shown the importance of omega-3 (ω-3) and omega-6 (ω-6) polyunsaturated fatty acids (PUFAs) for human and animal health. The long-chain eicosapentaenoic acid (EPA; 20:5ω-3) and docosahexaenoic acid (DHA; 22:6ω-3) are especially recognized for their nutritional value, and ability to alleviate many diseases in humans. So far, fish oil has been the main human source of EPA and DHA, but alternative sources are needed to satisfy the growing need for them. Therefore, we compared a fatty acid profile and content of 10 diatoms and seven dinoflagellates originating from marine, brackish and freshwater habitats. These two phytoplankton groups were chosen since they are excellent producers of EPA and DHA in aquatic food webs. Multivariate analysis revealed that, whereas the phytoplankton group (46%) explained most of the differences in the fatty acid profiles, habitat (31%) together with phytoplankton group (24%) explained differences in the fatty acid contents. In both diatoms and dinoflagellates, the total fatty acid concentrations and the ω-3 and ω-6 PUFAs were markedly higher in freshwater than in brackish or marine strains. Our results show that, even though the fatty acid profiles are genetically ordered, the fatty acid contents may vary greatly by habitat and affect the ω-3 and ω-6 availability in food webs.

## 1. Introduction 

Microalgae have been recognized as a source of functional ingredients for nutraceuticals with positive health effects, since they are able to synthetize many compounds essential for human nutrition [1,2]. Microalgae can be grown in controlled conditions, allowing the production of biomass with a constant biochemical composition and eliminating the risk of chemical contamination of the biomass. In the past few decades, the long-chain polyunsaturated omega-3 fatty acids (ω-3 PUFAs) have been recognized as compounds of great physiological importance for humans and other consumers [3]. In humans, eicosapentaenoic acid (EPA; 20:5ω-3) and docosahexaenoic acid (DHA; 22:6ω-3) are needed for the formation of eicosanoids, resolvins and protectins, which protect our cardiovascular system and alleviate e.g., renal diseases, inflammation, depression and tumor activity [4,5]. EPA and DHA are also essential for normal fetal brain development as well as growth and development of infants and children [3,4,5,6]. In aquatic ecosystems, the availability of ω-3 PUFAs has been related to egg production and hatching success of marine copepods [7,8] and freshwater cladocerans [9], as well as on the survival of fish larvae [10]. 

Most EPA and DHA are synthesized in aquatic environments, since terrestrial plants can only synthesize their precursors alpha-linolenic acid (ALA; 18:3ω-3) and stearidonic acid (SDA, 18:4ω-3) [2,11]. Some vertebrates can convert EPA and DHA from ALA and SDA through desaturation and elongation processes, but the conversion efficiency is generally low among vertebrates, including humans (e.g., in humans 1–4% [12,13]). Previous studies have shown that EPA and DHA synthesis is common only among some microalgal taxa [14,15,16]. Thus, the supply of the ω-3 PUFAs is dependent on the algal species composition in the aquatic food webs [16,17]. The amount of ω-3 PUFAs in the phytoplankton community is reflected to the upper trophic levels, ultimately affecting the nutritional quality of fish to humans [18,19]. 

In addition to ω-3 PUFAs, omega-6 (ω-6) PUFAs are also considered essential, but an excess dietary gain of ω-6 is related to negative health implications [5,20]. Excess ω-6, e.g., promote cardiovascular diseases, diabetes, cancer and inflammatory and autoimmune diseases in humans [20,21,22,23,24,25,26]. The main dietary sources of ω-6 PUFAs include plant oils such as sunflower, safflower, and corn oils, but they are also present in animal fat and wholegrain bread [5]. In the Western diet, the ω-6 PUFA consumption has become too high compared to ω-3 PUFAs due to increased consumption of linoleic acid (LA; 18:2ω-6) and arachidonic acid (ARA; 20:4ω-6), which are precursors of the long-chain ω-6 PUFAs [22,23]. The optimal dietary ω-6:3 -ratio is around 1 to 4:1, but in the Western diet the ratio varies between 10:1 and 20:1 [25]. This is significant, since ω-3 and ω-6 elongation uses the same enzymes, and thus in humans the excess consumption ω-6 fatty acids reduces the already low conversion rate of short chain ALA to long-chain EPA and DHA by 40–50% [26]. In some microalgae, the ω-6:3 -ratios are reported to be well below the suggested dietary ω-6:3 -ratio and thus microalgae could be especially suitable for products designed for balancing the dietary ω-6:3 -ratios [27]. 

Since EPA and DHA are primarily produced and transferred in the food chains of the aquatic environments, the demand for fish and seafood has constantly increased for filling the needs of the ever-growing human population [28]. Furthermore, over half of all fish and seafood is now being farmed and the only way to ensure high levels of EPA and DHA in farmed fish is to include these fatty acids in their feed [3,29]. Traditionally, this has been done by adding fish or fish oil into fish feed. Alternative sources are, however, needed because fish stock harvests cannot satisfy the consumer needs, and also the practice of using wild fish as feed at fish farms is unsustainable. As aquaculture and the required feed volumes have expanded, fish oil has been increasingly replaced by vegetable oils lacking EPA and DHA, but often containing high levels of ω-6 PUFAs [29]. This strategy has increased the sustainability of aquaculture, but impacted the nutritional quality of farmed fish that obtain less EPA and DHA and more ω-6 PUFAs than previously [30]. As a consequence, the amount of EPA and DHA has halved in farmed Atlantic salmon in 2006–2015 [31].

Based on previous studies, diatoms are known to be rich in EPA and dinoflagellates in DHA [14,15,16,17]. Both of them represent major groups of microalgae found in all kinds of aquatic habitats from oceans to brackish and freshwater environments, and their role in aquatic carbon and nutrient cycles is significant. Thus, they are both nutritionally important in different aquatic food webs and should offer great potential for commercial production of ω-3 PUFAs. However, in marine and brackish environments, some diatoms and dinoflagellates are known to be toxin producers, which obviously reduces their suitability for food and feed products [32]. Therefore, the commercial scale production of EPA and DHA from microalgae need to overcome several challenges starting from the screening and selection of strains [33]. 

Previous studies [14,15,16,17] have shown that the fatty acid profiles of algae are genetically determined, and, thus, when given as percentages of total fatty acids, more or less similar within taxonomical groups in marine and freshwater environments. However, fatty acid contents (as µg fatty acids per mg dry weight) are rarely reported, and thus it is not yet known whether the fatty acid contents resemble each other in the marine, brackish and freshwater strains. This can be misleading when considering the actual availability of the essential fatty acids in the food webs. Therefore, we cultured 10 diatoms and seven dinoflagellates of different habitats (marine, brackish, freshwater) and used multivariate statistical analysis to define the impact of habitat in fatty acid profiles and content in addition to phylogeny. In this study, we especially focused on comparing the ω-3 and ω-6 fatty acid content of diatoms and dinoflagellates originating from marine, brackish and freshwater habitats. This is the first attempt to study if the content of the bioactive ω-3 and ω-6 fatty acids varies systematically based on habitat. We did not study the toxicity of the strains, but—based on literature—the strains selected included both potentially toxic and non-toxic species.

We hypothesized that phylogeny is the main factor influencing the fatty acid profiles and contents of diatoms and dinoflagellates including the ω-3 and ω-6. We also selected three strain pairs, i.e., species that occurred in two of the three habitats, and expected them to have similar fatty acid profiles and contents irrespective of the habitat.

## 2. Results and Discussion

### 2.1. Growth Rates

The growth rate (µ) is one of the most important parameters when estimating the feasibility of commercial algal cultivation [33]. The maximal growth rates of both diatoms (0.07–0.46 divisions day^−1^) and dinoflagellates (0.08–0.33 divisions day^−1^) varied between strains, but could not be associated with the habitat, i.e., marine, brackish or freshwater (Table 1). The observed growth rates were in general in line with the ones reported for microalgae, e.g., for the diatoms *Chaetoceros brevis* and *Thalassiosira weissflogii* [34], the eustigmatophyte *Nannochloropsis,* and several other marine and freshwater species that all are considered as potential sources of commercial ω-3 PUFA production [35,36]. Trying to optimize the growth rates (i.e., adjusting the culture medium composition as well as salinity, temperature, and light conditions) was not within the scope of the present study, but, based on the observed growth in the current conditions, the marine strain of *Skeletonema marinoi* especially showed potential for commercial applications (Table 1). 

### 2.2. Phylogeny and Habitat Explaining the Proportional Fatty Acid Profiles and Contents of Diatoms and Dinoflagellates

Fatty acid profiles of diatoms and dinoflagellates varied by habitat, but also by strain (Figure 1). Principal component analysis (PCA) separated the diatoms and the dinoflagellates into their own groups in spite of the habitat (Figure 2A). In fact, the taxonomy explained 46% and habitat 13% of the variation in fatty acid profiles (PERMANOVA, Table 2). However, there was less dispersion among diatoms than among dinoflagellates in the PCA (Figure 2A) indicating stronger similarities in the fatty acid profiles of the studied diatoms than those of the dinoflagellates. Furthermore, the PERMANOVA also showed differences in fatty acid profiles of diatoms and dinoflagellates between the habitats. Habitat explained 39% (PERMANOVA, F_(2,20)_ = 5.82, *p* = 0.001) and 59% (PERMANOVA, F_(2,13)_ = 7.76, *p* = 0.001) of the variation in the proportional fatty acid profiles of diatoms and dinoflagellates, respectively. Pairwise PERMANOVA showed differences between brackish and marine (t = 2.20, *p* = 0.003) and freshwater (t = 3.14, *p* = 0.001) diatoms, whereas the fatty acid profiles of the marine and the freshwater diatoms did not differ (t = 1.44, *p* = 0.151). SIMPER analysis showed that these differences were related to a higher proportion of 16:1ω7c (explained 39–49% of differences) and 14:0 (explained 12–23% of differences) in brackish diatoms than in the marine or freshwater strains. In contrast to that, the brackish diatoms had a smaller proportion of EPA (explained 12–20% of differences) than the marine or the freshwater strains. The fatty acid profiles of brackish and freshwater dinoflagellates differed in oleic acid (18:1cis-9; explained 59% of differences) and in OPA (octadecapentaenoic acid; 18:5ω-3) (explained 12% of differences): the freshwater strains had more oleic acid and less OPA than the brackish strains. The comparison between the brackish and the freshwater or the marine strains could not be done since we succeeded to culture only one marine dinoflagellate. Galloway and Winder [16] has reported that the taxonomic group accounts for 3–4 times more variation in the fatty acid profiles than the environment; however, our study suggests that some taxonomic groups may not fit into this, and thus the habitat can explain more of the variation than suggested earlier.

Our PCA analysis on the fatty acid contents (µg FA mg in DW) of diatoms and dinoflagellates separated the strains into taxonomic groups by the component 2, whereas the component 1 separated the freshwater strains from the marine and the brackish (Figure 2B). Additionally, the freshwater diatoms were more positively and freshwater dinoflagellates more negatively related to component 1 than the brackish or marine strains. When all data were considered, the habitat explained 31% of the observed variation, whereas the taxonomic group explained only 24% (Table 2). More specifically, within diatoms the habitat explained 47% (PERMANOVA, F_(2,20)_ = 7.92, *p* = 0.001) of the variance in the fatty acid concentrations and in dinoflagellates 82% (PERMANOVA, F_(2,13)_ = 25.0, *p* = 0.001). The pairwise PERMANOVA showed difference (t = 2.5–3.1, *p* = 0.003–0.0011) between all habitats in diatoms and between brackish and freshwater dinoflagellates (t = 6.6, *p* = 0.001). The freshwater diatoms had higher EPA and 16:1ω7 content than the brackish or marine strains (SIMPER, EPA and 16:1ω7 explained 22% and 19% of dissimilarity, respectively), whereas the brackish strains had more 16:1ω7, 16:0 and 14:0 (explaining together 73% of dissimilarity) than the marine strains. This result shows that, even though the proportional fatty acid profiles are phylogenetically ordered, the actual fatty acid contents may vary depending on habitat.

### 2.3. The Total Fatty Acid Contents of Diatoms and Dinoflagellates

In addition to fatty acid profile and content of individual fatty acids, we compared total fatty acid content of diatoms and dinoflagellates. In the diatoms, the total sum of all fatty acids was highest in the freshwater strains and lowest in the marine strains (ANOVA *p* < 0.001; Table 3). The total fatty acid contents of the marine diatoms were actually surprisingly low (2.6–11.4 µg FA in mg DW) when compared to the concentrations reported earlier for microalgae (i.e., 12–40 µg FA in mg DW [43]). In the brackish diatom strains the total fatty acids (14.4–50.0 µg FA in mg DW) were close to the earlier reported [43], but in the freshwater strains substantially higher (53.8–133.9 µg FA in mg DW; Table 3). The nutritional value of diatoms for zooplankton has been shown to differ greatly at different stages of growth [7,44]. However, to take into account this variation, the microalgae were grown to the late exponential growth phase in this study. Thus, different growth phases should not be the cause for the obtained differences in the total fatty acid contents of the diatoms. The total fatty acid content did not differ between the two *S. marinoi* strains from the marine and brackish habitats, but was significantly higher in the freshwater than in the brackish *Diatoma tenuis* (ANOVA *p* < 0.001; Table 4).

A trend was observed in the total fatty acids of the dinoflagellates as well, but the order was freshwater > marine > brackish (Table 3). The total fatty acid concentrations in the marine (44.3 µg FA in mg DW) and the brackish dinoflagellates (27.1–83.3 µg FA in mg DW) were comparable to the earlier studies on microalgae (12–40 µg FA in mg DW [43]). As in the diatoms, also in the dinoflagellates, the total fatty acid concentrations were highest in the freshwater strains (136.5–143.0 µg FA in mg DW; Table 4). Among the dinoflagellates, the marine strain of *Apocalathium malmogiense* had a lower total fatty acid content than the brackish *A. malmogiense*, but this could not be tested due to the lack of sample replicates of the latter (Table 4). The detailed fatty acid profiles and concentrations (in µg FA in mg DW) of the studied diatom and dinoflagellate strains are shown in the Appendix A.

### 2.4. The w-3 and w-6 PUFA Contents and w-6:3 -Ratios of Diatoms and Dinoflagellates

The total sum of ω-3 and ω-6 PUFAs followed the trend of the total fatty acids. In general, the highest total ω-3 PUFA concentrations were detected in the freshwater strains (ANOVA *p* < 0.001; Table 3). The proportions of total ω-3 fatty acids of total fatty acids in the diatoms (range 4–39%) and in the dinoflagellates (range 19–49%; Figure 1) were within the range reported for diatoms and microalgae in general (8–31% [34,35,36]), but substantially lower than the values (range 64–81%) reported for marine cryptophytes [27]. The ω-3 profiles of the diatoms were more similar to each other than the profiles of the dinoflagellates from different habitats (Table 4). As expected, the diatoms had a higher EPA than DHA content and the dinoflagellates had a higher DHA than EPA content in all investigated habitats (Table 3 and Table 4). Even though the share of EPA in the diatoms was 3–35% of the total fatty acids, which is in accordance with the average proportions reported for marine and freshwater algae (10–25% and 5–20%, respectively) [2], the actual EPA concentrations were in general very low, and, in marine and brackish habitats, only slightly higher than in the dinoflagellates of the equivalent habitats. Among the diatoms, the highest total ω-3 concentrations (28.8 µg FA in mg DW) as well as EPA and DHA (23.1 and 2.4 µg FA in mg DW) concentrations were found in the freshwater *Nitzschia* sp. Marine cryptophytes are reported to be rich in ω-3, especially in EPA and DHA, and the total ω-3 and DHA values of the *Nitzschia* sp. were within the range detected from marine cryptophytes (ω-3 25–59 µg FA in mg DW, DHA 1-6 µg FA in mg DW) and the EPA contents were even higher than in marine cryptophytes (6–13 µg FA in mg DW) [27]. In the dinoflagellates, the EPA content was up to ten times higher in the studied strains originating from freshwater habitats compared to those from marine and brackish habitats, and a similar trend was found also in DHA although the differences were smaller (Table 3 and Table 4). Due to the differences in the total fatty acid concentrations, the proportions of DHA of total fatty acids were similar in the three habitats, i.e., 14–27%, which are higher than the average proportions reported for marine and freshwater microalgae (5–10% and 1–3%, respectively) [2]. The highest DHA concentrations were detected in the freshwater dinoflagellate *Gymnodinium fuscum* (Table 4). These concentrations were ten times higher than those measured from marine cryptophytes [27]. 

The ALA and SDA contents were low both in the diatoms (1–6% of total fatty acids) and in the dinoflagellates (1–13% of total fatty acids), in all studied habitats (Figure 1 and Figure 3, Table 3). These values are in accordance with the results of Twining and co-workers [2], who found that ALA is more commonly available in terrestrial (15–35% of the total fatty acids of the primary producers) than in aquatic habitats (<15% of the total fatty acids of the primary producers) [2]. The marine diatom *S. marinoi* had a higher total ω-3 and especially EPA content than the brackish *S. marinoi*, and the freshwater diatom *D. tenuis* had a higher total ω-3, ALA, SDA and EPA content than the brackish *D. tenuis* (Table 4). Furthermore, the marine and brackish dinoflagellate *A. malmogiense* differed in ω-3 fatty acids, the brackish strain having higher concentrations than the marine strain (Table 4). Our observations of the lower ω-3 PUFA concentrations in the studied marine and brackish diatoms and dinoflagellates than in the freshwater strains indicate that there may be less ω-3 PUFAs available in saline food webs than in freshwater food webs (Figure 3). All in all, the studied marine and brackish strains showed low potential for commercial EPA and DHA production.

Diatoms and dinoflagellates are important bloom-formers in both marine and brackish habitats, and among them many toxin producing species occur [32,37,38,39,40,41,42]. Contrarily, in freshwater, toxic blooms of diatoms or dinoflagellates are rarely reported and toxic blooms are typically caused by cyanobacteria [38]. Of the studied species, the diatom *Pseudo-nitzschia pungens* and the dinoflagellates *Alexandrium ostenfeldii* and *Karlodinium veneficum* are known toxin producers [40,41,42]. The toxin production was not within the scope of this study, but we are also aware that of ω-3 fatty acids, OPA excreted e.g., by some marine gymnodinioid dinoflagellate species, has been claimed to cause toxic effects e.g., in sea urchins [45], trout hepatocytes [46] and sheep red blood cells [47]. However, opposite results have also been published, and the current opinion is that OPA is not causing the toxicity [48]. It has also been reported that phytoplankton synthesize either OPA or EPA but rarely both [17]. Small amounts (1-7 µg FA in mg DW) of OPA were found in all studied marine and brackish dinoflagellates; however, the brackish *A. malmogiense* had a substantially higher OPA content (23 µg FA in mg DW). In diatoms, OPA was found only in minor amounts (0.02 µg FA in mg DW) and only in *Melosira arctica* and both *S. marinoi* strains (Table 4). Interestingly, the highest amount (17.5 µg FA in mg DW) of OPA was detected in the freshwater *G. fuscum*, which among dinoflagellates also had the highest EPA content (13.7 µg FA in mg DW) (Table 4). 

The ω-6 PUFA concentrations were more or less similar in all environments (<2.6 µg FA in mg DW; Table 3 and Table 4), and the amounts were lower than those reported for e.g., cryptophytes and brown algae [27,49]. The ω-6 concentrations were also very low compared to ω-3 PUFAs, and constituted only 1–4% of the total fatty acids (Figure 1, Table 3). When comparing the species occurring in two habitats, higher total ω-6 PUFA concentrations were found in the brackish than in the marine dinoflagellate *A. malmogiense* and in the freshwater than in the brackish diatom *D. tenuis* (ANOVA *p* < 0.001). The marine and brackish *S. marinoi* did not differ in their total ω-6 content (Table 4). Similarly to total ω-6, the LA and ARA concentrations also did not vary between the habitats (Table 3 and Table 4). However, the ARA concentration in the freshwater *Nitzschia* sp. was significantly higher than in any other studied strain (ANOVA *p* < 0.01; Table 4). DPA was detected only in one strain, i.e., in the freshwater diatom *Stephanodiscus hantzschii*, which had minor DPA concentration (Table 4). Small amounts (0.01–0.1 µg FA in mg DW) of gamma-linolenic acid (GLA; 18:3ω-6) were found in all studied strains except for the freshwater diatoms, the marine diatom *Thalassiosira nordenskioldii* and the freshwater dinoflagellate *G. fuscum* (Table 4). The ω-6:3 -ratios varied a lot and were more or less similar in the diatoms and the dinoflagellates irrespective of the habitat (Table 4). The lowest ω-6:3 -ratios were detected in the diatom *Nitzschia* sp. (1:95) and in the dinoflagellate *Peridinium cinctum* (1:68), but also, in all other strains, the ω-6:3 -ratios were low and thus, in that sense, potentially suitable for products designed for balancing the ω-6:3 -ratios of the Western diet [20].

We cannot explain the clear differences in the total fatty acid concentrations or in the specific ω-3 fatty acids between the saline and freshwater strains. It is known that the growth conditions (nutrients, light and temperature) affect the fatty acid contents of microalgae. Regarding the nutrients, nitrogen limitation is often linked to high fatty acid production [17]. The f/2 and MWC media are not very different in their nitrogen content, but the phosphorus content is higher in MWC than in f/2 [50,51]. The availability of phosphorus is reported to affect the fatty acid contents of algae, but the direction of the effect varies on genus level, and, for example, in diatoms, the phosphorus limitation increases the amount of total lipids, but decreases the PUFA content [17]. For testing the effects of nutrients, temperature and light, we should have cultured all strains in all conditions, including in culturing the saline strains in freshwater medium and vice versa, which would most probably have been impossible, and was also out of the scope of our study. 

## 3. Materials and Methods

### 3.1. Algal Culturing and Growth Rate Determinations

The microalgal strains were acquired from culture collections (Table 1). We investigated altogether 17 strains. The starting point was the habitat of intermediate salinity, i.e., the brackish environment, and the strains were selected based on them being common and/or bloom-forming taxa and thus of ecosystem relevance in the northern Baltic Sea (60°N 26°E). These species were as far as possible paired with marine and freshwater counterparts, selecting, whenever available, taxa of the same species, the same genus, or the same order. The selection process was affected by the availability of the strains in the culture collections, and the different character of the habitats, and ultimately our success in culturing them, thus not all strains had counterparts in all habitats. However, the selected strains included three strain pairs, i.e., species that originated in two different habitats. These were the diatoms *Diatoma tenuis* (brackish and freshwater strains) and *Skeletonema marinoi* (marine and brackish strains), and the dinoflagellate *Apocalathium malmogiense* (marine and brackish strains). The strains also included both potentially toxic and non-toxic species (Table 1).

All brackish water strains originated from the Baltic Sea. The brackish diatoms were cultured in F⁄2 medium [50] and the brackish dinoflagellates in F/2-Si, both based on 6 psu filtered (0.2 μm) and autoclaved Baltic Sea seawater (FSW). The marine diatoms were likewise cultured in F/2 and the marine dinoflagellates in F/2-Si, both based on 30 psu FSW prepared using Tropic Marin^®^ PRO-REEF artificial sea salt (Tropic Marin AG, Hünenberg, Switzerland) and the above mentioned Baltic Sea FSW. The culturing was done under a daily 16 h light: 8 h dark cycle at a light level of 115 µmol quanta s^−1^ m^−2^. The temperature varied according to strain, but followed the temperatures given for each strain by the culture collections (Table 1). The strains were cultured in 550 mL plastic tissue culture flasks, five flasks per strain, giving a total of 2 L culture per strain. Two flasks per strain were sampled regularly (every two to seven days) to attain cell densities for growth rate calculation. The freshwater strains were cultured in MWC-medium [51] in 600 mL plastic tissue culture flasks. Each strain had two replicates, which were grown at 18 °C and under a 16 h light: 8 h dark cycle at a light level of 70 µmol quanta s^−1^ m^−2^. The culture conditions were not specifically optimized for growth or fatty acid production, but followed the instructions provided by the culture collections. The freshwater diatoms were sampled for cell number determinations every second day, but the growth rates were not determined for the freshwater dinoflagellates. Depending on culture, it took 15–30 days to attain the desired growth phase and cell density.

For growth rate calculations, the samples were preserved with acid Lugol’s solution [52] and cell densities were counted using a Sedgewick–Rafter chamber (Paul Marienfeld GmbH & Co. KG, Lauda-Königshofen, Germany). The specific growth rates (µ; divisions day^−1^) for all strains were calculated for the exponential growth phase using Equation 1: µ = ln(cellsTx/cellsT0).(1)

### 3.2. Total Lipid Extraction, Fatty Acid Transesterification and Analysis by GC-MS

The samples for fatty acid analyses were collected by centrifugation (2000 rpm for 10 min) in the late exponential growth phase. The obtained pellets were placed into −80 °C and freeze dried within a week. For the analysis, the total lipids were extracted from the freeze-dried samples no longer than four weeks after the sampling. Due to the low amount of biomass, we obtained only one sample from the brackish dinoflagellate *A. malmogiense*. Otherwise, two replicates of homogenized biomass samples (1–3 mg) were extracted using chloroform:methanol 2:1 (by volume). The samples were sonicated for 10 min to maximize extraction results, after which the samples were vortexed, and centrifuged for 5 min at 2000 rpm. Toluene and 1% sulfuric acid in methanol were used for the transesterification of fatty acid methyl esters (FAME). We used 1,2-dinonadecanoyl-sn-glycero-3-phosphatidylcholine (Larodan, Solna, Sweden) and tricosanoic acid (Nu-Chek Prep Inc., Elysian, MN, USA) as internal standards. The samples were heated (90 °C for 90 min), after which they were neutralized with 1.5 mL of 2% of KHCO_3_ and diluted in 2 mL hexane. The tubes were vortexed and centrifuged (2 min at 1500 rpm) and the upper layer was collected for the analysis. 

FAMEs were analyzed with a gas chromatograph (Shimadzu Ultra, Kyoto, Japan) equipped with mass detector (GC-MS) and using helium as a carrier gas (linear velocity = 36.3 cm s^−1^). Temperature of injector was 270 °C and we used splitless injection mode (for 1 min). Temperatures of interface and ion source were 250 °C and 220 °C, respectively. Phenomenex^®^ (Torrance, CA, USA) ZB-FAME column (30 m × 0.25 mm × 0.20 µm) with 5 m Guardian was used with the following temperature program: 50 °C was maintained for 1 min, then the temperature was increased at 10 °C min^−1^ to 130 °C, followed by 7 °C min^−1^ to 180 °C, and 2 °C min^−1^ to 200 °C and held for 3 min, and finally heated at 10 °C min^−1^ to 260 °C. Total program time was 35.14 min and solvent cut time 9 min. Fatty acids were identified by the retention times (RT) and using specific ions [18], which were also used for quantification. Fatty acid concentrations were calculated using calibration curves based on known standard solutions (15 ng, 50 ng, 100 ng and 250 ng) of a FAME standard mixture (GLC standard mixture 566c, Nu-Chek Prep, Elysian, MN, USA) and using recovery percentage of internal standards. The Pearson correlation coefficient was >0.99 for each individual fatty acid calibration curve. 

### 3.3. Statistics

The differences in the fatty acid contents (total fatty acids, ω-3, combined ALA and SDA, OPA, EPA, DHA, ω-6, LA, GLA, ARA, DPA) between the microalgal strains and habitats (marine, brackish and freshwater) were studied with ANOVA and Tukey’s honestly significant difference (HSD) post hoc test. Levene’s test was used for testing the homogeneity of variances. We used principal component analysis (PCA) ordinations for data visualization. Since PCA is not a statistical test, and the ANOVA tests only for the similarity of the group averages, we used PERMANOVA (permutational multivariate analysis of variance) and SIMPER (similarity percentages) for a more detailed study of the similarity of the fatty acid concentrations and proportional profiles of the diatoms and the dinoflagellates in the three habitats. PERMANOVA was run with unrestricted permutation of raw data and type III sums of squares. In statistical testing, *p*-values < 0.05 were considered as statistically significant. The statistical analyses were performed using IBM SPSS Statistics 22 (SPSS Inc., Chicago, IL, USA), and the PCA, PERMANOVA and SIMPER were done using Primer 7 (version 7.0.13, Quest Research Limited, Auckland, New Zealand). 

## 4. Conclusions

We hypothesized that phylogeny is the main factor influencing the fatty acid profiles and contents of diatoms and dinoflagellates including the ω-3 and ω-6. However, this was true only partly: when the fatty acid profiles were considered, the proportional differences in the fatty acids were explained by taxonomy, whereas the actual fatty acid concentrations were more related to the habitat than to phylogeny. Based on the strains studied, the fatty acid concentrations were systematically highest in the strains originating from freshwater. We also found that the proportional fatty acid profiles of the marine and the brackish diatoms and dinoflagellates were more similar to each other than to their freshwater counterparts. Nevertheless, the diatoms always had higher EPA than DHA and the dinoflagellates higher DHA than EPA concentrations, irrespective of the habitat. From the studied three strain pairs, we found significant habitat-related differences in *Diatoma tenuis* and *Apocalathium malmogiense*. Altogether, our results imply that the freshwater food webs may be richer in ω-3 PUFAs than the marine and brackish food webs. However, more data is needed to draw definite conclusions.

The most promising diatom strain for commercial EPA production turned out to be the freshwater *Nitzschia* sp., whereas the highest DHA content was detected in the freshwater dinoflagellate *Gymnodinium fuscum*. This dinoflagellate also had a high total fatty acid and total ω-3 PUFA content. Based on their ω-6:3 -ratios, all of the studied strains are potentially suitable for products targeted to lower the ω-6:3 -ratios of the Western diet. We did not study the toxicity of the strains but are aware that many microalgal species are known to produce toxins. Thus, as with all novel food and feed products, the safety of any microalgal strain needs to be guaranteed and legally certified before possible commercialization. 

## Figures and Tables

**Figure 1 marinedrugs-17-00233-f001:**
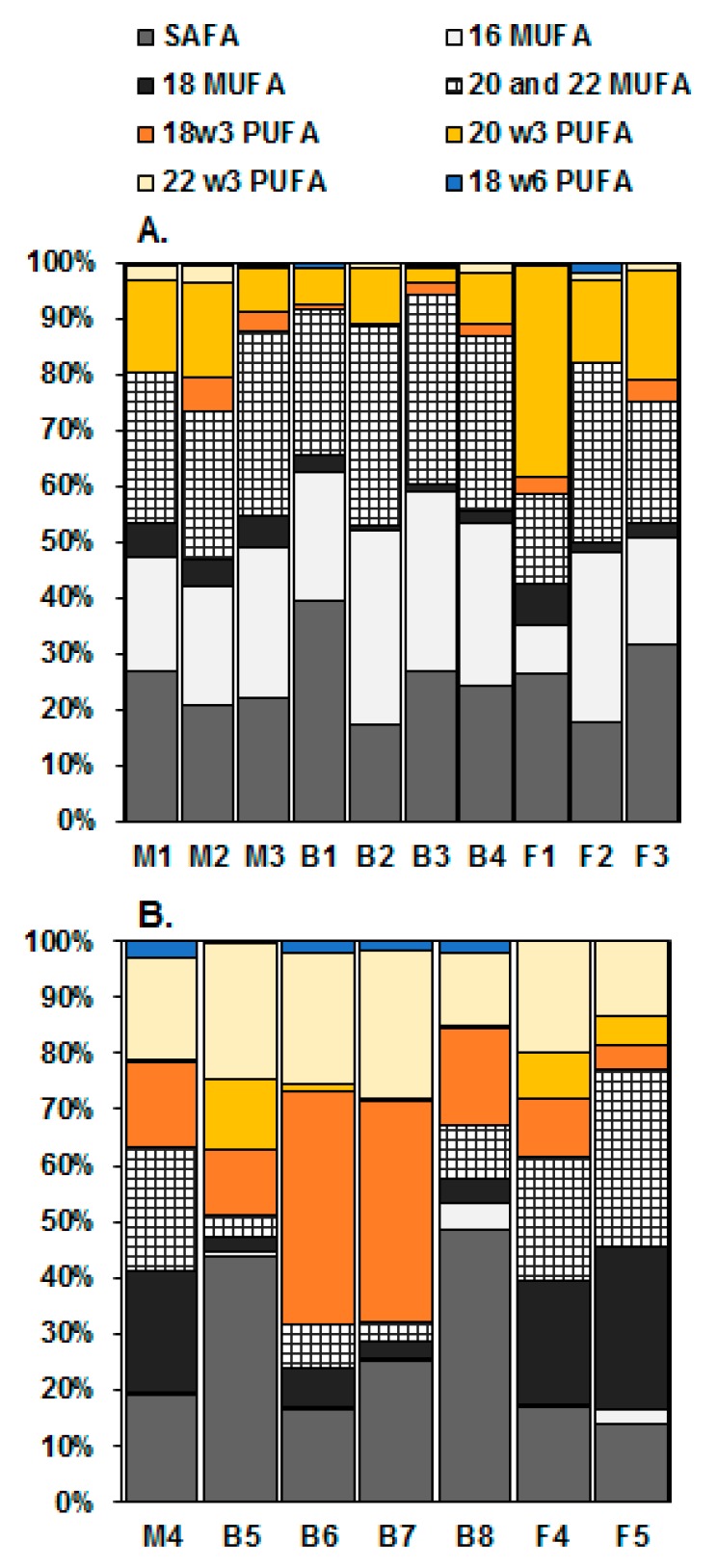
The fatty acid profiles of the studied diatoms (**A**) and dinoflagellates (**B**) as % of the total fatty acids. 18ω-3 is the sum of alpha-linolenic acid (ALA), stearidonic acid (SDA) and octadecapentaenoic acid (OPA), 20ω-3 is eicosapentaenoic acid (EPA), 22ω-3 is docosahexaenoic acid (DHA), 18ω-6 is the sum of linoleic acid (LA) and gamma-linolenic acid (GLA), MUFA is mono-unsaturated fatty acids (16 MUFA, 18 MUFA and other MUFA) and SAFA is saturated fatty acids. For numbering of strains, see Table 1.

**Figure 2 marinedrugs-17-00233-f002:**
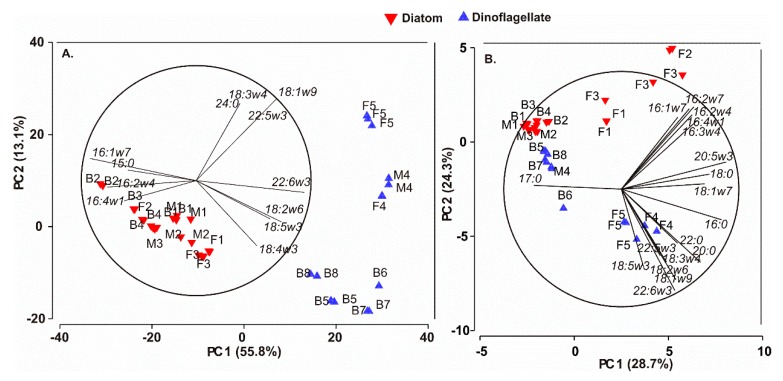
The principal component analysis (PCA) plot of the studied marine, brackish and freshwater diatoms (red inverted triangles) and dinoflagellates (blue triangles) based on (**A**) their fatty acid profiles (as % of the total fatty acids), and (**B**) based on their fatty acid concentrations (µg FA in mg DW). The marine strains are denoted with the letter M, brackish with B and freshwater with F; for numbering of strains, see Table 1. The strain pairs, i.e., species that occurred in two of the three habitats, are M2-B3, B1-F1, and M4-B6.

**Figure 3 marinedrugs-17-00233-f003:**
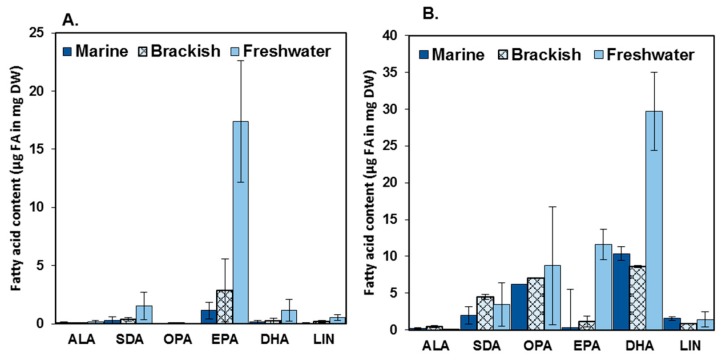
The concentrations (µg FA in mg DW) of the most abundant ω-3 and -6 fatty acids ALA (alpha-linolenic acid), SDA (stearidonic acid), OPA (octadecapentaenoic acid), EPA (eicosapentaenoic acid), DHA (docosahexaenoic acid) and LA (linoleic acid) of diatoms (**A**) and dinoflagellates (**B**) in the three studied habitats.

**Table 1 marinedrugs-17-00233-t001:** The investigated marine, brackish and freshwater diatom and dinoflagellate strains and their growth conditions (medium, temperature, and light intensity) as well as their potential toxicity based on literature [37,38,39,40,41,42], the cell numbers in the cultures in the late exponential growth phase, and maximal growth rates (calculated for the exponential growth phase using Equation 1). Strain numbers refer to Figure 1 and Figure 2.

Habitat	Number	Diatoms	Origin of Strain	Collection Site, Isolated by (Year)	Medium	Temperature (°C)	Light (µmol quanta s^−1^ m^−2^)	Potentially Toxic (Yes/No)	Cells (cells mL^−1^)	µ Max (divisions day^−1^)
**Marine**	M1	*Pseudo-nitzschia pungens* CCAP 1061/44	CCAP	LY1, Argyll, Scotland, UK; Garvetto (2015)	F/2+Si, 30 psu	16	115	yes	5.5 × 10^4^	0.18
	M2	*Skeletonema marinoi* K-0669	NORCCA (SCCAP)	Hvidøre, Øresund (The Sound), Denmark; N.H. Larsen (2004)	F/2+Si, 30 psu	4	115	no	1.3 × 10^6^	0.46
	M3	*Thalassiosira nordenskioldii* CCAP 1085/31	CCAP	Arctic waters, West Greenland Shelf; Fragoso (2013)	F/2+Si, 30 psu	4	115	no	3.7 × 10^4^	0.22
**Brackish**	B1	*Diatoma tenuis* DTTV B5	FINMARI CC	Tvärminne Storfjärden, Baltic Sea; A. Kremp (2008)	F/2+Si, 6 psu	4	115	no	1.0 × 10^5^	0.33
	B2	*Melosira arctica* MATV-1402	FINMARI CC	Längden, Tvärminne, Baltic Sea; J. Oja (2014)	F/2+Si, 6 psu	4	115	no	2.1 × 10^5^	0.32
	B3	*Skeletonema marinoi* Skeletonema	FINMARI CC	Tvärminne Storfjärden, Baltic Sea; K. Spilling (2006)	F/2+Si, 6 psu	4	115	no	1.1 × 10^6^	0.25
	B4	*Thalassiosira baltica* TBLL7-1301	FINMARI CC	LL7, Gulf of Finland, Baltic Sea; A. Kremp (2013)	F/2+Si, 6 psu	4	115	no	7.2 × 10^4^	0.19
**Freshwater**	F1	*Diatoma tenuis* CPCC 62	CPCC	Lake Ontario, Canada; C. Ewing (1984)	MWC	18	70	no	2.6 × 10^4^	0.22
	F2	*Nitzschia* sp. FD397	UTEX	Minnesota, USA; D. Czarnecki (1998)	MWC	18	70	no	1.5 × 10^5^	0.20
	F3	*Stephanodiscus hantzschii* CPCC 267 (CCAP 1079/4)	CPCC	Esthwaite Water, Cumbria, England, UK; Jaworski (1983)	MWC	18	70	no	6.9 × 10^5^	0.07
		**Dinoflagellates**								
**Marine**	M4	*Apocalathium malmogiense* K-0399	NORCCA (SCCAP)	Igloolik Island, Canada; N. Daugbjerg (1989)	F/2-Si, 30 psu	4	115	no	1.3 × 10^4^	0.12
**Brackish**	B5	*Alexandrium ostenfeldii* AOF-0926	FINMARI CC	Föglö, Åland Islands, Baltic Sea; A. Kremp (2009)	F/2-Si, 6 psu	16	115	yes	3.9 × 10^3^	0.08
	B6	*Apocalathium malmogiense* SHTV-5	FINMARI CC	Tvärminne Storfjärden, Baltic Sea; A. Kremp (2002)	F/2-Si, 6 psu	4	115	no	2.6 × 10^4^	0.14
	B7	*Heterocapsa triquetra* HTF 1002	FINMARI CC	Föglö, Åland Islands, Baltic Sea; P. Hakanen (2010)	F/2-Si, 6 psu	16	115	no	3.0 × 10^4^	0.33
	B8	*Karlodinium veneficum* KVDAN 22	FINMARI CC	Danskog, Raseborg, Baltic Sea; M. Parrow (2013)	F/2-Si, 6 psu	16	115	yes	5.0 × 10^4^	0.16
**Freshwater**	F4	*Gymnodinium fuscum* K-1836	NORCCA (SCCAP)	Lake Fiolen, Sweden; G. Hansen (2012)	MWC	18	70	no	1.3 × 10^6^	nd
	F5	*Peridinium cinctum* K-1721	NORCCA (SCCAP)	Freshwater rockpool, Tvärminne, Finland; G. Hansen (2011)	MWC	18	70	no	2.6 × 10^4^	nd

Abbreviations: CCAP = Culture Collection of Algae and Protozoa; FINMARI CC = FINMARI Culture Collection/SYKE Marine Research Centre and Tvärminne Zoological Station; NORCCA = Norwegian Culture Collection of Algae; SCCAP = (the former) Scandinavian Culture Collection for Algae & Protozoa; UTEX = UTEX Culture Collection of Algae at the University of Texas at Austin. nd = not determined.

**Table 2 marinedrugs-17-00233-t002:** Permutational multivariate analysis of variance (PERMANOVA) results for comparisons of fatty acid concentration by phytoplankton group, i.e., taxonomy (Ph), habitat (Ha), and their interactions. SS sum of squares, MS mean squares, P(perm) significance, P(MC) significance after Montecarlo correction.

Source	df	SS	MS	Pseudo-F	P (perm)	P (MC)
Profile Model						
Ph	1	13771	13771	53.987	0.001	0.001
Ha	2	3846	1923	7.5405	0.001	0.001
PhxHa	2	3310	1655	6.4893	0.001	0.001
Res	29	7397	255			
Total	34	29649				
Content Model						
Ph	1	6260	6260	30.196	0.001	0.001
Ha	2	7956	3978	19.186	0.001	0.001
PhxHa	2	5159	2579	12.442	0.001	0.001
Res	29	6012	207			
Total	34	26083				

**Table 3 marinedrugs-17-00233-t003:** Habitat-specific diatom and dinoflagellate averages of total fatty acids, ω-3, ALA, SDA, OPA, EPA, DHA, ω-6, LA, GLA, ARA, DPA and ω-6:3 ratios. Standard deviations are given in the parenthesis.

Taxa	Habitat	Total FA (µg FA mg DW^−1^)	ω-3 (µg FA mg DW^−1^)	ALA (µg FA mg DW^−1^)	SDA (µg FA mg DW^−1^)	OPA (µg FA mg DW^−1^)	EPA (µg FA mg DW^−1^)	DHA (µg FA mg DW^−1^)	ω-6 (µg FA mg DW^−1^)	LA (µg FA mg DW^−1^)	GLA (µg FA mg DW^−1^)	ARA (µg FA mg DW^−1^)	DPA (µg FA mg DW^−1^)	ω-6:3 -ratio
**Diatoms**	Marine	7.6 (4.5)	1.7 (1.2)	0.1 (0.1)	0.3 (0.3)	0.01 (0.01)	1.1 (0.7)	0.2 (0.1)	0.1 (0.04)	0.1 (0.04)	0.007 (0.0007)	0.003 (0.003)	-	1:25
	Brackish	26.3 (15.2)	3.7 (3.2)	0.02 (0.01)	0.4 (0.2)	0.01 (0.01)	2.9 (2.7)	0.3 (0.2)	0.3 (0.2)	0.2 (0.1)	0.04 (0.02) *	0.1 (0.1)	-	1:14
	Freshwater	88.9 (36.5) *	21.1 (6.2) *	0.2 (0.1) *	1.5 (1.2)	-	17.4 (5.2) *	1.2 (0.9) *	0.6 (0.3)	0.5 (0.3)	-	0.9 (0.8)	0.1 (0.1) *	1:36
**Dinoflagellates**	Marine	44.3 (1.1) ^†^	13.7 (0.8) ^†^	0.2 (0.01) ^†^	2.0 (0.1) ^†^	6.3 (0.2) ^†^	0.3 (0.1) ^†^	10.4 (0.6) ^†^	1.7 (0.1) ^†^	1.6 (0.04) ^†^	0.12 (0.01) ^†^	0.3 (0.005) ^†^	-	1:8 ^†^
	Brackish	37.8 (20.3)	15.1 (10.4)	0.4 (0.3)	4.5 (4.3)	7.0 (7.0)	1.1 (1.0)	8.6 (5.7)	1.0 (0.6)	0.9 (0.5)	0.04 (0.002)	0.3 (0.3)	-	1:16
	Freshwater	140.4 (7.0) *	45.8 (5.3) *	0.1 (0.1)	3.4 (2.9)	8.7 (8.0)	11.6 (2.1) *	29.7 (5.3) *	1.4 (1.0)	1.4 (1.0)	0.03 (0.005)	0.02 (0.02)	-	1:32

Abbreviations: Total FA (total fatty acids), ω-3 (total ω-3 fatty acids), ALA + SDA (alpha-linolenic acid, 18:3ω−3 and stearidonic acid, 18:4:ω−3), OPA (octadecapentaenoic acid; 18:5ω−3), EPA (eicosapentaenoic acid, 20:5ω−3), DHA (docosahexaenoic acid, 22:6ω−3), ω-6 (total ω-6 fatty acids), LA (linoleic acid, 18:2ω−6), GLA (gamma-linolenic acid; 18:3ω−6), ARA (arachidonic acid, 20:4ω−6), DPA (docosapentaenoic acid, 22:5ω−6). The statistical testing was done with ANOVA, * = *p* < 0.01. † = the results of dinoflagellates could not be statistically tested due to the low number of marine strains.

**Table 4 marinedrugs-17-00233-t004:** Strain-specific averages of total fatty acids, ω-3, ALA, SDA, OPA, EPA, DHA, ω-6, LA, GLA, ARA, DPA and ω-6:3 ratios of the studied strains from the three studied habitats, i.e., marine, brackish and freshwater environments. Standard deviations are given in the parenthesis.

Habitat	Diatoms	Total FA (µg FA mg DW^−1^)	ω-3 (µg FA mg DW^−1^)	ALA (µg FA mg DW^−1^)	SDA (µg FA mg DW^−1^)	OPA (µg FA mg DW^−1^)	EPA (µg FA mg DW^−1^)	DHA (µg FA mg DW^−1^)	ω-6 (µg FA mg DW^−1^)	LA (µg FA mg DW^−1^)	GLA (µg FA mg DW^−1^)	ARA (µg FA mg DW^−1^)	DPA (µg FA mg DW^−1^)	ω-6:3 -ratio
**Marine**	*Pseudo-nitzschia pungens* CCAP 1061/44	2.6 (1.2)	0.6 (0.2)	0.0003 (0.001)	-	-	0.5 (0.02)	0.08 (0.03)	0.03 (0.03)	0.03 (0.03)	0.01 (0.001)	0.02 (0.01)	-	1:24
	*Skeletonema marinoi* K-0669	11.4 (3.4)	2.9 (0.6)	0.03 (0.005)	0.7 (0.2)	0.02 (0.02)	1.9 (0.4)	0.3 (0.02)	0.1 (0.01)	0.1 (0.01)	0.01 (0.001)	-	-	1:27
	*Thalassiosira nordenskioldii* CCAP 1085/31	8.8 (2.5)	1.4 (0.9)	0.2 (0.1)	0.2 (0.1)	-	1.0 (0.7)	0.1 (0.05)	0.1 (0.02)	0.1 (0.1)	-	-	-	1:24
**Brackish**	*Diatoma tenuis* DTTV B5	16.8 (2.1)	1.7 (0.5)	0.01 (0.01)	0.2 (0.1)	-	1.3 (0.5)	0.02 (0.04)	0.3 (0.1)	0.2 (0.02)	0.05 (0.01)	0.01 (0.002)	-	1:5
	*Melosira arctica* MATV-1402	50.0 (3.7) *	8.5 (0.4) *	0.03 (0.01)	0.3 (0.01)	0.02 (0.002)	7.0 (0.3) *	0.5 (0.002) *	0.5 (0.01)	0.3 (0.006)	0.09 (0.002)	0.4 (0.0003)	-	1:18
	*Skeletonema marinoi* Skeletonema	14.4 (0.1)	1.0 (0.1)	0.02 (0.001)	0.4 (0.002)	0.02 (0.05)	0.5 (0.1)	0.1 (0.05)	0.1 (0.002)	0.1 (0.01)	0.01 (0.003)	0.007 (0.001)	-	1:9
	*Thalassiosira baltica* TBLL7-1301	24.1 (0.7)	3.8 (0.1)	0.4 (0.1)	0.6 (0.03)	-	2.7 (0.1)	0.4 (0.01)	0.1 (0.01)	0.1 (0.01)	0.02 (0.004)	0.01 (0.001)	-	1:29
**Freshwater**	*Diatoma tenuis* CPCC 62	53.8 (0.4) *	20.7 (0.2) *	0.3 (0.01)	1.2 (0.02)	-	18.9 (0.3) *	0.2 (0.01)	0.9 (0.04) *	0.9 (0.02) *	-	0.2 (0.01)	-	1:23
	*Nitzschia* sp. FD397	133.9 (2.5) *	28.8 (0.2) *	0.03 (0.01)	0.3 (0.01)	-	23.1 (0.2) *	2.4 (0.03) *	0.3 (0.03)	0.3 (0.01)	-	3.0 (0.4) *	-	1:95
	*Stephanodiscus hantzschii* CPCC 267 (CCAP 1079/4)	82.3 (26.0) *	16.3 (4.8) *	0.2 (0.1)	2.6 (1.0)	-	12.6 (3.5) *	1.0 (0.2) *	0.6 (0.2)	0.4 (0.1)	-	0.03 (0.01)	0.1 (0.03) *	1:29
	**Dinoflagellates**													
**Marine**	*Apocalathium malmogiense* K-0399	44.3 (1.1)	13.4 (0.8)	0.2 (0.1)	2.0 (0.1)	6.2 (0.3)	0.3 (0.1)	10.4 (0.6)	1.7 (0.1)	1.6 (0.01)	0.1 (0.01)	0.3 (0.01)	-	1:8
**Brackish**	*Alexandrium ostenfeldii* AOF-0926	27.1 (0.7)	13.4 (0.6)	0.1 (0.01)	3.0 (0.1)	1.1 (0.003)	3.4 (0.02)	6.6 (0.08)	0.9 (0.04)	0.8 (0.03)	0.04 (0.004)	0.03 (0.0003)	-	1:14
	*Apocalathium malmogiense* SHTV-5	83.3 ^†^	36.8 ^†^	0.5 ^†^	13.3 ^†^	23.0 ^†^	0.8 ^†^	20.6 ^†^	2.3 ^†^	1.9 ^†^	1.1 ^†^	0.2 ^†^	-	1:16
	*Heterocapsa triquetra* HTF 1002	32.9 (2.1)	15.5 (1.2)	0.9 (0.3)	5.1 (1.4)	7.3 (0.6)	0.1 (0.01)	9.0 (0.7)	0.5 (0.01)	0.5 (0.001)	0.02 (0.001)	0.3 (0.01)	-	1:29
	*Karlodinium veneficum* KVDAN 22	30.7 (3.0)	5.9 (1.1)	0.3 (0.04)	0.9 (0.1)	4.7 (0.8)	0.2 (0.01)	4.3 (0.8)	0.7 (0.1)	0.7 (0.02)	0.02 (0.002)	0.2 (0.01)	-	1:8
**Freshwater**	*Gymnodinium fuscum* K-1836	136.5 (6.6) *	48.1 (3.8) *	0.02 (0.02)	0.2 (0.01)	17.5 (0.1) *	13.7 (0.2) *	34.1 (1.1) *	2.6 (0.1) *	2.2 (0.1) *	-	0.05 (0.004)	-	1:19
	*Peridinium cinctum* K-1721	143.0 (8.1) *	44.3 (6.9) *	0.1 (0.01) *	5.6 (0.5) *	2.9 (0.4)	10.2 (1.3) *	26.8 (4.8) *	0.7 (0.05)	0.7 (0.01) *	0.06 (0.01)	-	-	1:68

Abbreaviations: total FA (total fatty acids), ω-3 (total ω-3 fatty acids, ALA + SDA (alpha-linolenic acid, 18:3ω−3 and stearidonic acid, 18:4:ω−3), OPA (octadecapentaenoic acid; 18:5ω−3), EPA (eicosapentaenoic acid, 20:5ω−3), DHA (docosahexaenoic acid, 22:6ω−3), ω-6 (total ω-6 fatty acids), LA (linoleic acid, 18:2ω−6), GLA (gamma-linolenic acid; 18:3ω−6), ARA (arachidonic acid, 20:4ω−6), DPA (docosapentaenoic acid, 22:5ω−6). The statistical testing was done with ANOVA, * = *p* < 0.01. † = the results of the marine *A. malmogiense* could not be statistically tested due to a low number of replicates from this strain.

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
