# Peer review of "Comparison of Diatoms and Dinoflagellates from Different Habitats as Sources of PUFAs"

_marinedrugs, 2019, doi:10.3390/md17040233_

Round 1
Reviewer 1 Report
The introduction is well done and poses the problem of reducing the amounts of omega 3 in farmed fish by adding vegetable oil in their food.
However, more work is needed on the justification of the choice of strains in relation to the literature. This could be done in a table. In the same way, it is better to justify the choice of growing conditions (temperature, light). Are these conditions already optimized in the literature for the production of lipids for these strains?
And finally, I will have rejected in this choice the three potentially toxic strains.
This work on the choice of strains, will allow to the authors to improve their discussion, which is a little short for the moment.
I do not understand the ratio notation at the line 66 : 1 to 4 :1 or 1.4 :1 ?
In Result and Discussion, is it possible to have the lipid rate in Table 1? Or put the level of total fatty acid? This, to have a comparison between the number of cells and the rate of lipid?
Author Response
Reviewer 1:
The introduction is well done and poses the problem of reducing the amounts of omega 3 in farmed fish by adding vegetable oil in their food.
However, more work is needed on the justification of the choice of strains in relation to the literature. This could be done in a table. In the same way, it is better to justify the choice of growing conditions (temperature, light). Are these conditions already optimized in the literature for the production of lipids for these strains?
Thank you for your valuable comments. We have reviewed the manuscript accordingly. The biggest changes are highlighted with yellow.
Trying to optimize the growth rates (i.e. adjusting the culture medium composition as well as salinity, temperature, and light conditions) was not within the scope of the present study. Regarding the temperatures, the growth conditions were selected as far as possible based on the recommendations given by the culture collections, regarding the light, the 16:8 L:D cycle is commonly used for microalgae, and also the used light quanta of 70 and 115 µmol s-1 m-2 are within normal in microalgal cultivation. However, the ideal growth conditions may vary between the strains, and were not optimized for each strain or specifically for fatty acid production. This is now explained in the text. Further, we have added some discussion on the reason for the higher fatty acid contents in freshwater strains related to growth conditions. We also found a small error in our fatty acid content calculations, and this is now corrected. However, the error did not affect the proportional results and increased all concentrations more or less equally.
We have also added justification about the selection process of the strains: “The starting point was the habitat of intermediate salinity, i.e. the brackish environment, and the strains were selected based on them being common and/or bloom-forming taxa and thus of ecosystem relevance in the northern Baltic Sea (60°N 26°E). These species were as far as possible paired with marine and freshwater counterparts, selecting, whenever available, taxa of the same species, the same genus, or the same order. The selection process was affect by the availability of the strains in the culture collections, and the different character of the habitats, and ultimately our success in culturing them, thus not all strains had counterparts in all habitats.”
And finally, I will have rejected in this choice the three potentially toxic strains.
We understand this comment, and it is very relevant in terms of commercial fatty acid production. However, our aim was not only to seek for commercial solutions, but also to compare the different aquatic habitats. Thus, we find it important to include the results of these three strains even though they might be toxic. We did not study the toxicity of the strains, but in Table 1 point out that some of the species that we studying are known to have toxic strains. We also point out that we are aware that many microalgal species are known to produce toxins, and that as with all novel food and feed products, the safety of any microalgal strain needs to be guaranteed and legally certified before possible commercialization.
This work on the choice of strains, will allow to the authors to improve their discussion, which is a little short for the moment.
Based on the comments from the other referees, we have done additional statistical tests and improved the results and discussion accordingly.
I do not understand the ratio notation at the line 66 : 1 to 4 :1 or 1.4 :1 ?
It is 1 to 4:1, this is now clarified in the text.
In Result and Discussion, is it possible to have the lipid rate in Table 1? Or put the level of total fatty acid? This, to have a comparison between the number of cells and the rate of lipid?
The authors do not understand what is meant here with the “lipid rate”. Total lipids were not studied here (that would include fatty acids, carotenoids and some other pigments as well as sterols). The total fatty acids are already given in Table 3 for each strain. However, the fatty acid content is not dependent on the number of cells, since it is given as µg fatty acids in mg dry weight. Thus, the comparison between the number of cells and the rate of lipid cannot be done.
Reviewer 2 Report
This manuscript report omega 3 fatty acid conents from 17 algal strains from marine, barckish and freshwater habitats, suggesting freshwater diatoms and dinoflagellates are better candidates in omega 3 fatty acid production. I think that their approach comparing different habitats are quite interesting but has potentials to mislead readers. To be published, the authors should answer to several comments of mine.
line 152-154: It appears that freshwater diatoms and dinoflagellates have higher total fatty acid contnent, which make higher conten of EPA and DHA. Why do they have higher FA content? Please search and present other literature explaining why they have such higher total FA content. Also, please supply relevent literature regarding total fatty acid content (in terms of micro g FA per DW) ranges between marine and freshwater species in Results and Discussion.
line 331-334: The authors stats that those strains examined in their study have higher FA concentrations without showing literature values. Please show literature values of EPA and DHA in terms of micro g FA per DW for other species in marine and freshwater environment. A recent review by Twining et al. (2016) showed much higher average EPA and DHA concentration (in terms of % total fatty acids) in marine species. I agree that for omega 3 fatty acid production, FA content in terms of micro g per DW. Without presenting data for much more species, the manuscript might misleading readers to conclude that freshwater species in general might be superior to marine/brackish one in omega 3 FA content.
line 340-341: Based on my comments above, the authors' statement appear to be somewhat risky. I would rather remove the sentence.
references
Twining et al. 2016. Highly unsaturated fatty acids in nature: what we know and what we need to learn. Oikos 125: 749-760.
Author Response
Reviewer 2:
This manuscript report omega 3 fatty acid conents from 17 algal strains from marine, barckish and freshwater habitats, suggesting freshwater diatoms and dinoflagellates are better candidates in omega 3 fatty acid production. I think that their approach comparing different habitats are quite interesting but has potentials to mislead readers. To be published, the authors should answer to several comments of mine.
line 152-154: It appears that freshwater diatoms and dinoflagellates have higher total fatty acid contnent, which make higher conten of EPA and DHA. Why do they have higher FA content? Please search and present other literature explaining why they have such higher total FA content. Also, please supply relevent literature regarding total fatty acid content (in terms of micro g FA per DW) ranges between marine and freshwater species in Results and Discussion.
Thank you for your comments. We have reviewed the manuscript accordingly. The biggest changes are highlighted with yellow.
The discussion and literature on the high FA content of freshwater strains is now added. (Please, see also our response to the next comment regarding the fatty acids contents in µg FA in mg DW). The following is now added into the text “We cannot explain the clear differences in the total fatty acid concentrations or in the specific ω-3 fatty acids between the saline and freshwater strains. It is known that the growth conditions (nutrients, light and temperature) affect the fatty acid contents of microalgae. Regarding the nutrients, nitrogen limitation is often linked to high fatty acid production [17]. The f/2 and MWC media are not very different in their nitrogen content, but the phosphorus content is higher in MWC than in f/2 [50,51]. The availability of phosphorus is reported to affect the fatty acid contents of algae, but the direction of the effect varies on genus level, and for example in diatoms the phosphorus limitation increases the amount of total lipids, but decreases the PUFA content [17]. For testing the effects of nutrients, temperature and light, we should have cultured all strains in all conditions, including in culturing the saline strains in freshwater medium and vice versa, which would most probably have been impossible, and was also out of the scope of our study.” We also found a small error in our fatty acid content calculations, and this is now corrected. However, the error did not affect the proportional results and increased all concentrations more or less equally.
line 331-334: The authors stats that those strains examined in their study have higher FA concentrations without showing literature values. Please show literature values of EPA and DHA in terms of micro g FA per DW for other species in marine and freshwater environment. A recent review by Twining et al. (2016) showed much higher average EPA and DHA concentration (in terms of % total fatty acids) in marine species. I agree that for omega 3 fatty acid production, FA content in terms of micro g per DW. Without presenting data for much more species, the manuscript might misleading readers to conclude that freshwater species in general might be superior to marine/brackish one in omega 3 FA content.
Thank you for this comment. We have now added discussion and literature on EPA and DHA concentrations in algae. However, fatty acid results are surprisingly rarely given in the literature as FA per DW, and the percentage values used by many authors are actually not very informative when the total amount of fatty acids are not given. This is also now discussed in the text.
line 340-341: Based on my comments above, the authors' statement appear to be somewhat risky. I would rather remove the sentence.
The sentence “both the diatom and the dinoflagellate strains from this habitat had higher concentrations than those previously reported for microalgae” is now removed from lines 340-341.
Reviewer 3 Report
Dear authors
In this manuscript (marinedrugs-477960), Peltomaa et al. present the omega-3 fatty acid composition of diatoms and dinoflagellates from marine, freshwater and brackish environmental. Overall the manuscript is well written, has novelty and present relevant results. However, in my opinion, the authors don’t draw from the results obtained the full potential they have.
The introduction is well documented, the results presentation and results discussion should be improved, while the materials and methods are not complete. If the authors address in a revised version the points listed below, they will get a much best quality manuscript.
1. The authors do not emphasize what is really new in this work. In fact, who reads the manuscript gets the idea that the results are in accordance with the described in the literature and nothing else. Both in the abstract and in the conclusions and at each point in the text, the authors should clearly show the reader which result is new and what implications entail that they are innovative in relation to what is described in the literature.
2. If the authors carry out the analysis of omega-3 and omega-6 fatty acids, why do not they reflect this in the title of the manuscript? As shown in table 2 and 3, the authors analyse the content of EPA and DHA, and also ARA and LA, which mean omega-3 and omega 6 and its ratio. Thus, sentences like line 94 makes no sense. I suggest the authors change the text to show they determine and discuss the omega-3 and omega-6 fatty acid composition.
3. Table 1, 2 and 3, are very small and difficult to read, while the table title is too long. Please, construct the tables according to the journal instructions. The title should be short and explanatory of the table contents but should not be a detailed explanation of the content of each column. For example, the units of each variable in each column must be included in the header of each column. Part of the title information should be transformed into a table note (eg the meaning of abbreviations).
4. Authors should deepen the discussion of results. The data presented in Tables 2 and 3 goes well beyond the DHA and EPA content. Therefore, they should not be limited to discuss the factors related to the production of these two fatty acids. For example, they explore very little from PCA analysis. Authors should get more information from PCA analysis. For example, PC1 is the component that accounts for almost 61% of all variances between samples, and PC1 being responsible for the large difference between F4 and F5, and between these and F1 and F2, the authors should indicate which fatty acids that most contribute to this differentiation of the samples.
5. Several points indicated in the attached file show that the experimental methods are not described in the essential details. For example, the authors should present the experimental conditions used in GC-MS analysis such as the temperature program, as well as describe how they performed the quantitative analysis (what standards did they use, what range of concentrations did they use in the analysis, what is the calibration line obtained ...)
6. The conclusions are too long. Please provide only the main conclusions. The aim of the work is not conclusions.
7. Title: all words should start with capital letter
8. The final list of references should be carefully reviewed. Authors should be consistent by carefully following the formatting indicated in the journal's instructions. For example, the name of the journal should be abbreviated using points in each abbreviated word and in italic, the year should be in bold, the volume must be in italics… Please follow the instructions (Author 1, A.B.; Author 2, C.D. Title of the article. Abbreviated Journal Name Year, Volume, page range.
9. Several other minor points are assigned in the pdf attached file. Please address them also.

Author Response
Reviewer 3:
Dear authors
In this manuscript (marinedrugs-477960), Peltomaa et al. present the omega-3 fatty acid composition of diatoms and dinoflagellates from marine, freshwater and brackish environmental. Overall the manuscript is well written, has novelty and present relevant results. However, in my opinion, the authors don’t draw from the results obtained the full potential they have.
The introduction is well documented, the results presentation and results discussion should be improved, while the materials and methods are not complete. If the authors address in a revised version the points listed below, they will get a much best quality manuscript.
1. The authors do not emphasize what is really new in this work. In fact, who reads the manuscript gets the idea that the results are in accordance with the described in the literature and nothing else. Both in the abstract and in the conclusions and at each point in the text, the authors should clearly show the reader which result is new and what implications entail that they are innovative in relation to what is described in the literature.
Thank you for your constructive comments, they helped improve the manuscript. The biggest changes are highlighted with yellow.
We have modified the text and emphasized our findings in order for them to better be distinguished from the literature references.
2. If the authors carry out the analysis of omega-3 and omega-6 fatty acids, why do not they reflect this in the title of the manuscript? As shown in table 2 and 3, the authors analyse the content of EPA and DHA, and also ARA and LA, which mean omega-3 and omega 6 and its ratio. Thus, sentences like line 94 makes no sense. I suggest the authors change the text to show they determine and discuss the omega-3 and omega-6 fatty acid composition.
The omega-6 in now added into the title of the manuscript, and both omega-3 and omega-6 are now discussed more clearly in the text. We have edited the PCA-plots to cover all fatty acids and done additional statistical tests with PERMANOVA and SIMPER in order to show more details of the fatty acid profiles and omega-3 and -6 of the studied strains. We also found a small error in our fatty acid content calculations, and this is now corrected. However, the error did not affect the proportional results and increased all concentrations more or less equally.
3. Table 1, 2 and 3, are very small and difficult to read, while the table title is too long. Please, construct the tables according to the journal instructions. The title should be short and explanatory of the table contents but should not be a detailed explanation of the content of each column. For example, the units of each variable in each column must be included in the header of each column. Part of the title information should be transformed into a table note (eg the meaning of abbreviations).
We are sorry about this inconvenience with the tables, the correct-sized tables were included in the submission, but for some reason were not visible for the reviewers. The units are now added into each column and the abbreviations are now explained as table notes. All tables can now be found in the end of the document.
4. Authors should deepen the discussion of results. The data presented in Tables 2 and 3 goes well beyond the DHA and EPA content. Therefore, they should not be limited to discuss the factors related to the production of these two fatty acids. For example, they explore very little from PCA analysis. Authors should get more information from PCA analysis. For example, PC1 is the component that accounts for almost 61% of all variances between samples, and PC1 being responsible for the large difference between F4 and F5, and between these and F1 and F2, the authors should indicate which fatty acids that most contribute to this differentiation of the samples.
Thank you for this comment. Since EPA and DHA are nutritionally most valuable and desired fatty acids, we concentrated on them. However, we have now added discussion on other omega-3 and omega-6 fatty acids as well.
5. Several points indicated in the attached file show that the experimental methods are not described in the essential details. For example, the authors should present the experimental conditions used in GC-MS analysis such as the temperature program, as well as describe how they performed the quantitative analysis (what standards did they use, what range of concentrations did they use in the analysis, what is the calibration line obtained ...)
This information on GC-MS analysis is now given in the methods.
6. The conclusions are too long. Please provide only the main conclusions. The aim of the work is not conclusions.
The conclusions are now edited.
7. Title: all words should start with capital letter
The title is now edited.
8. The final list of references should be carefully reviewed. Authors should be consistent by carefully following the formatting indicated in the journal's instructions. For example, the name of the journal should be abbreviated using points in each abbreviated word and in italic, the year should be in bold, the volume must be in italics… Please follow the instructions (Author 1, A.B.; Author 2, C.D. Title of the article. Abbreviated Journal Name Year, Volume, page range.
These are now edited.
9. Several other minor points are assigned in the pdf attached file. Please address them also.
These are now edited.